# Distributional Reward Decomposition for Reinforcement Learning

**Zichuan Lin**[*]
Tsinghua University
linzc16@mails.tsinghua.edu.cn

**Li Zhao**
Microsoft Research
lizo@microsoft.com

**Derek Yang**
UC San Diego
dyang1206@gmail.com

**Tao Qin**
Microsoft Research
taoqin@microsoft.com

**Guangwen Yang**
Tsinghua University
ygw@tsinghua.edu.cn

**Tie-Yan Liu**
Microsoft Research
tyliu@microsoft.com

## Abstract

Many reinforcement learning (RL) tasks have specific properties that can be leveraged to modify existing RL algorithms to adapt to those tasks and further improve performance, and a general class of such properties is the multiple reward channel. In those environments the full reward can be decomposed into sub-rewards obtained from different channels. Existing work on reward decomposition either requires prior knowledge of the environment to decompose the full reward, or decomposes reward without prior knowledge but with degraded performance. In this paper, we propose Distributional Reward Decomposition for Reinforcement Learning (DRDRL), a novel reward decomposition algorithm which captures the multiple reward channel structure under distributional setting. Empirically, our method captures the multi-channel structure and discovers meaningful reward decomposition, without any requirements on prior knowledge. Consequently, our agent achieves better performance than existing methods on environments with multiple reward channels.

## 1 Introduction

Reinforcement learning has achieved great success in decision making problems since Deep Q-learning was proposed by Mnih et al. [2015]. While general RL algorithms have been deeply studied, here we focus on those RL tasks with specific properties that can be utilized to modify general RL algorithms to achieve better performance. Specifically, we focus on RL environments with multiple reward channels, but only the full reward is available.

Reward decomposition has been proposed to investigate such properties. For example, in Atari game *Seaquest*, rewards of environment can be decomposed into sub-rewards of shooting sharks and those of rescuing divers. Reward decomposition views the total reward as the sum of sub-rewards that are usually disentangled and can be obtained independently (Sprague and Ballard [2003], Russell and Zimdars [2003], Van Seijen et al. [2017], Grimm and Singh [2019]), and aims at decomposing the total reward into sub-rewards. The sub-rewards may further be leveraged to learn better policies.

Van Seijen et al. [2017] propose to split a state into different sub-states, each with a sub-reward obtained by training a general value function, and learn multiple value functions with sub-rewards. The architecture is rather limited due to requiring prior knowledge of how to split into sub-states. Grimm and Singh [2019] propose a more general method for reward decomposition via maximizing

---

[*]Contributed during internship at Microsoft Research.

disentanglement between sub-rewards. In their work, an explicit reward decomposition is learned via maximizing the disentanglement of two sub-rewards estimated with action-value functions. However, their work requires that the environment can be reset to arbitrary state and can not apply to general RL settings where states can hardly be revisited. Furthermore, despite the meaningful reward decomposition they achieved, they fail to utilize the reward decomposition into learning better policies.

In this paper, we propose Distributional Reward Decomposition for Reinforcement Learning (DR-DRL), an RL algorithm that captures the latent multiple-channel structure for reward, under the setting of distributional RL. Distributional RL differs from value-based RL in that it estimates the distribution rather than the expectation of returns, and therefore captures richer information than value-based RL. We propose an RL algorithm that estimates distributions of the sub-returns, and combine the sub-returns to get the distribution of the total returns. In order to avoid naive decomposition such as 0-1 or half-half, we further propose a disentanglement regularization term to encourage the sub-returns to be diverged. To better separate reward channels, we also design our network to learn different state representations for different channels.

We test our algorithm on chosen Atari Games with multiple reward channels. Empirically, our method has following achievements:

- Discovers meaningful reward decomposition.
- Requires no external information.
- Achieves better performance than existing RL methods.

## 2 Background

We consider a general reinforcement learning setting, in which the interaction of the agent and the environment can be viewed as a Markov Decision Process (MDP). Denote the state space by $\mathcal{X}$, action space by $A$, the state transition function by $P$, the action-state dependent reward function by $R$ and $\gamma$ the discount factor, we write this MDP as $(\mathcal{X}, A, R, P, \gamma)$.

Given a fixed policy $\pi$, reinforcement learning estimates the action-value function of $\pi$, defined by $Q^{\pi}(x, a) = \sum_{t=0}^{\infty} \gamma^t r_t(x_t, a_t)$ where $(x_t, a_t)$ is the state-action pair at time $t$, $x_0 = x, a_0 = a$ and $r_t$ is the corresponding reward. The Bellman equation characterizes the action-value function by temporal equivalence, given by

$$Q^{\pi}(x, a) = R(x, a) + \gamma \mathop{\mathbb{E}}_{x', a'} [Q^{\pi}(x', a')]$$

where $x' \sim P(\cdot|x, a), a' \sim \pi(\cdot|x')$. To maximize the total return given by $\mathop{\mathbb{E}}_{x_0, a_0} [Q^{\pi}(x_0, a_0)]$, one common approach is to find the fixed point for the Bellman optimality operator

$$Q^*(x, a) = \mathcal{T} Q^*(x, a) = R(x, a) + \gamma \mathop{\mathbb{E}}_{x'} \left[ \max_{a'} Q^*(x', a') \right]$$

with the temporal difference (TD) error, given by

$$\delta_t^2 = \left[ r_t + \gamma \max_{a' \in \mathcal{A}} Q(x_{t+1}, a') - Q(x_t, a_t) \right]^2$$

over the samples $(x_t, a_t, s_t, x_{t+1})$ along the trajectory. Mnih et al. [2015] propose Deep Q-Networks (DQN) that learns the action-value function with a neural network and achieves human-level performance on the Atari-57 benchmark.

### 2.1 Reward Decomposition

Studies for reward decomposition also leads to state decomposition (Laversanne-Finot et al. [2018], Thomas et al. [2017]), where state decomposition is leveraged to learn different policies. Extending their work, Grimm and Singh [2019] explore the decomposition of the reward function directly, which is considered to be most related to our work. Denote the $i$-th ($i$=1,2,...,$N$) sub-reward function at state-action pair $(x, a)$ as $r_i(x, a)$, the complete reward function is given by

$$r = \sum_{i=1}^{N} r_i$$

For each sub-reward function, consider the sub-value function $U_i^\pi$ and corresponding policy $\pi_i$:

$$U_i^\pi(x_0, a_0) = \mathbb{E}_{x_t, a_t}[\sum_{t=0}^{\infty} \gamma^t r_i(x_t, a_t)]$$

$$\pi_i = \arg\max_\pi U_i^\pi$$

where $x_t \sim P(\cdot|\pi, x_0, a_0), a_t \sim \pi(\cdot|x_t)$.

In their work, reward decomposition is considered meaningful if each reward is obtained independently (i.e. $\pi_i$ should not obtain $r_j$) and each reward is obtainable.

Two evaluate the two desiderata, the work proposes the following values:

$$J_{independent}(r_1, \ldots, r_n) = \mathbb{E}_{s\sim\mu}\left[\sum_{i\neq j} \alpha_{i,j}(s)U_i^{\pi_j^*}(s)\right], \tag{1}$$

$$J_{nontrivial}(r_1, \ldots, r_n) = \mathbb{E}_{s\sim\mu}\left[\sum_{i=1}^{n} \alpha_{i,i}(s)U_i^{\pi_i^*}(s)\right], \tag{2}$$

where $\alpha_{i,j}$ is for weight control and for simplicity set to 1 in their work. During training, the network would maximize $J_{nontrivial} - J_{independent}$ to achieve the desired reward decomposition.

## 2.2 Distributional Reinforcement Learning

In most reinforcement learning settings, the environment is not deterministic. Moreover, in general people train RL models with an $\epsilon$-greedy policy to allow exploration, making the agent also stochastic. To better analyze the randomness under this setting, Bellemare et al. [2017] propose C51 algorithm and conduct theoretical analysis of distributional RL.

In distributional RL, reward $R_t$ is viewed as a random variable, and the total return is defined by $Z = \sum_{t=0}^{\infty} \gamma^t R_t$. The expectation of $Z$ is the traditional action-value $Q$ and the distributional Bellman optimality operator is given by

$$\mathcal{T}Z(x, a) :\overset{D}{=} R(x, a) + \gamma Z\left(x', \arg\max_{a'\in\mathcal{A}} \mathbb{E}Z(x', a')\right)$$

where if random variable $A$ and $B$ satisfies $A \overset{D}{=} B$ then $A$ and $B$ follow the same distribution.

Random variable is characterized by a categorical distribution over a fixed set of values in C51, and it outperforms all previous variants of DQN on Atari domain.

# 3 Distributional Reward Decomposition for Reinforcement Learning

## 3.1 Distributional Reward Decomposition

In many reinforcement learning environments, there are multiple sources for an agent to receive reward as shown in Figure 1(b). Our method is mainly designed for environments with such property.

Under distributional setting, we will assume reward and sub-rewards are random variables and denote them by $R$ and $R_i$ respectively. In our architecture, the categorical distribution of each sub-return $Z_i = \sum_{t=0}^{\infty} \gamma^t R_{i,t}$ is the output of a network, denoted by $\mathcal{F}_i(x, a)$. Note that in most cases, sub-returns are not independent, i.e. $P(Z_i = v)! = P(Z_i = v|Z_j)$. So theoretically we need $\mathcal{F}_{ij}(x, a)$ for each $i$ and $j$ to obtain the distribution of the full return. We call this architecture as non-factorial model or full-distribution model. The non-factorial model architecture is shown in appendix. However, experiment shows that using an approximation form of $P(Z_i = v) \approx P(Z_i = v|Z_j)$ so that only $\mathcal{F}_i(x, a)$ is required performs much better than brutally computing $\mathcal{F}_{ij}(x, a)$ for all $i, j$, we believe that is due to the increased sample number. In this paper, we approximate the conditional probability $P(Z_i = v|Z_j)$ with $P(Z_i = v)$.

Consider categorical distribution function $\mathcal{F}_i$ and $\mathcal{F}_j$ with same number of atoms $K$, the $k$-th atom is denoted by $a_k$ with value $a_k = a_0 + kl, 1 \leq k \leq K$ where $l$ is a constant. Let random variable

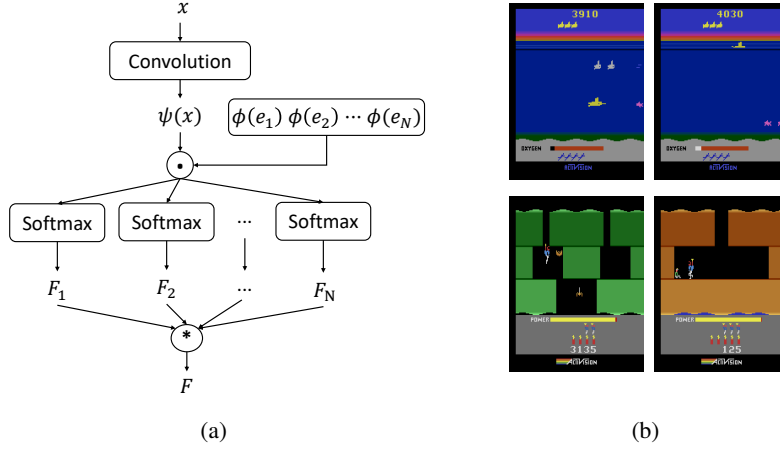

(a)                                        (b)

Figure 1: (a) Distributional reward decomposition network architecture. (b) Examples of multiple reward channels in Atari games: the top row shows examples of *Seaquest* in which the submarine receives rewards from both shooting sharks and rescuing divers; the bottom row shows examples of *Hero* where the hero receives rewards from both shooting bats and rescuing people.

$Z_i \sim \mathcal{F}_i$ and $Z_j \sim \mathcal{F}_j$, from basic probability theory we know that the distribution function of $Z = Z_i + Z_j$ is the convolution of $F_i$ and $F_j$

$$
\begin{aligned}
\mathcal{F}(v) = P(Z_i + Z_j = v) &= \sum_{k=1}^{K} P(Z_i = a_k) P(Z_j = v - a_k | Z_i = a_k) \\
&\approx \sum_{k=1}^{K} P(Z_i = a_k) P(Z_j = v - a_k) = \mathcal{F}_i(v) * \mathcal{F}_j(v).
\end{aligned}
\tag{3}
$$

When we use $N$ sub-returns, the distribution function of the total return is then given by $\mathcal{F} = \mathcal{F}_1 * \mathcal{F}_2 * \cdots * \mathcal{F}_N$ where $*$ denotes linear 1D-convolution.

While reward decomposition is not explicitly done in our algorithm, we can derive the decomposed reward with using trained agents. Recall that total return $Z = \sum_{i=1}^{N} Z_i$ follows bellman equation, so naturally we have

$$
\mathcal{T}Z \stackrel{D}{=} \mathcal{T}(\sum_{i=1}^{N} Z_i) \stackrel{D}{=} R + \gamma Z' = (\sum_{i=1}^{N} R_i) + \gamma(\sum_{i=1}^{N} Z_i')
\tag{4}
$$

where $Z_i'$ represents sub-return on the next state-action pair. Note that we only have access to a sample of the full reward $R$, the sub-rewards $R_i$ are arbitrary and for better visualization a direct way of deriving them is given by

$$
R_i = Z_i - \gamma Z_i'
\tag{5}
$$

In the next section we will present an example of those sub-rewards by taking their expectation $\mathbb{E}(R_i)$. Note that our reward decomposition is latent and we do not need $R_i$ for our algorithm, Eq. 5 only provides an approach to visualize our reward decomposition.

## 3.2 Disentangled Sub-returns

To obtain meaningful reward decomposition, we want the sub-rewards to be disentangled. Inspired by Grimm and Singh [2019], we compute the disentanglement of distributions of two sub-returns $F^i$ and $F^j$ on state $x$ with the following value:

$$
J_{disentang}^{ij} = D_{KL}(\mathcal{F}_{x, \arg\max_a \mathbb{E}(Z_i)} || \mathcal{F}_{x, \arg\max_a \mathbb{E}(Z_j)}),
\tag{6}
$$

where $D_{KL}$ denotes the cross-entropy term of KL divergence.

Intuitively, $J_{disentang}^{ij}$ estimates the disentanglement of sub-returns $Z_i$ and $Z_j$ by first obtaining actions that maximize $\mathbb{E}(Z_i)$ and $\mathbb{E}(Z_j)$ respectively, and then computes the KL divergence between the two estimated total returns of the actions. If $Z_i$ and $Z_j$ are independent, the action maximizing two sub-returns would be different and such difference would reflect in the estimation for total return. Through maximizing this value, we can expect a meaningful reward decomposition that learns independent rewards.

## 3.3 Projected Bellman Update with Regularization

Following the work of C51 (Bellemare et al. [2017]), we use projected Bellman update for our algorithm. When applied with the Bellman optimality operator, the atoms of $\mathcal{T}Z$ is shifted by $r_t$ and shrank by $\gamma$. However to compute the loss, usually KL divergence between $Z$ and $\mathcal{T}Z$, it is required that the two categorical distributions are defined on the same set of atoms, so the target distribution $\mathcal{T}Z$ would need to be projected to the original set of atoms before Bellman update. Consider a sample transition $(x, a, r, x')$, the projection operator $\Phi$ proposed in C51 is given by

$$(\Phi \mathcal{T} Z(x,a))_i = \sum_{j=0}^{M-1} \left[ 1 - \frac{\left| [r + \gamma a_j]_{V_{min}}^{V_{max}} - a_i \right|}{l} \right]_0^1 \mathcal{F}_{x',a'}(a_j), \tag{7}$$

where $M$ is the number of atoms in C51 and $[\cdot]_a^b$ bounds its argument in $[a,b]$. The sample loss for $(x, a, r, x')$ would be given by the cross-entropy term of KL divergence of $Z$ and $\Phi \mathcal{T} Z$

$$\mathcal{L}_{x,a,r,x'} = D_{KL}(\Phi \mathcal{T} Z(x,a) || Z(x,a)). \tag{8}$$

Let $\mathcal{F}^\theta$ be a neural network parameterized by $\theta$, we combine distributional TD error and disentanglement to jointly update $\theta$. For each sample transition $(x, a, r, x')$, $\theta$ is updated by minimizing the following objective function:

$$\mathcal{L}_{x,a,r,x'} - \lambda \sum_i \sum_{j!=i} J_{disentang}^{ij}, \tag{9}$$

where $\lambda$ denotes learning rate.

## 3.4 Multi-channel State Representation

One complication of our approach outlined above is that very often the distribution $\mathcal{F}_i$ cannot distinguish itself from other distributions (e.g., $\mathcal{F}_j, j \neq i$) during learning since they all depend on the same state feature input. This brings difficulties in maximizing disentanglement by jointly training as different distribution functions are exchangeable. A naive idea is to split the state feature $\psi(x)$ into $N$ pieces (e.g., $\psi(x)_1, \psi(x)_2, ..., \psi(x)_N$) so that each distribution depends on different sub-state-features. However, we empirically found that this method is not enough to help learn good disentangled sub-returns.

To address this problem, we utilize an idea similar to universal value function approximation (UVFA) (Schaul et al. [2015]). The key idea is to take one-hot embedding as an additional input to condition the categorical distribution function, and apply the element-wise multiplication $\psi \odot \phi$, to force interaction between state features and the one-hot embedding feature:

$$\mathcal{F}_i(x,a) = \mathcal{F}_{\theta_i}(\psi(x) \odot \phi(e_i))_a, \tag{10}$$

where $e_i$ denotes the one-hot embedding where the $i$-th element is one and $\phi$ denotes one-layer non-linear neural network that is updated by backpropagation during training.

In this way, the agent explicitly learns different distribution functions for different channels. The complete network architecture is shown in Figure 1(a).

## 4 Experiment Results

We tested our algorithm on the games from Arcade Learning Environment (ALE; Bellemare et al. [2013]). We conduct experiments on six Atari games, some with complicated rules and some with

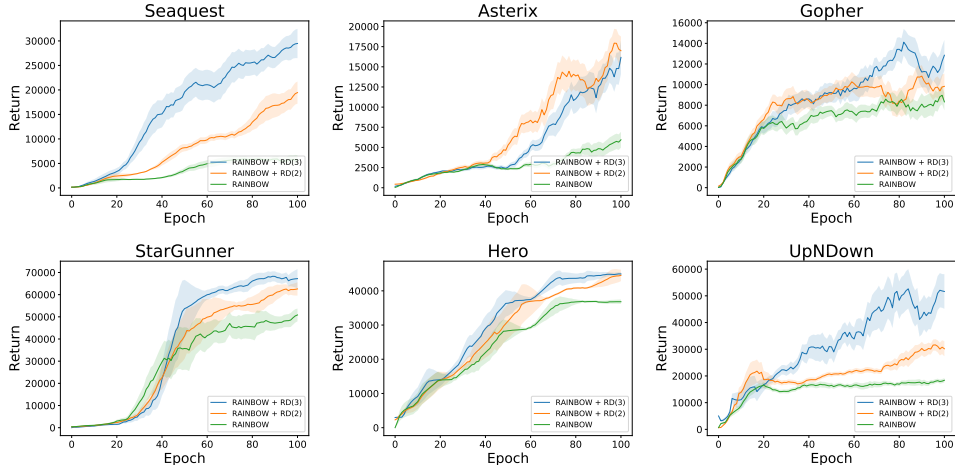

Figure 2: Performance comparison with Rainbow. RD(N) represents using N-channel reward decomposition. Each training curve is averaged by three random seeds.

simple rules. For our study, we implemented our algorithm based on Rainbow (Hessel et al. [2018]) which is an advanced variant of C51 (Bellemare et al. [2017]) and achieved state-of-the-art results in Atari games domain. We replace the update rule of Rainbow by Eq. 9 and network architecture of Rainbow by our convoluted architecture as shown in Figure 1(a). In Rainbow, the Q-value is bounded by $[V_{min}, V_{max}]$ where $V_{max} = -V_{min} = 10$. In our method, we bound the categorical distribution of each sub-return $Z_i(i = 1, 2, ..., N)$ by a range of $[\frac{V_{min}}{N}, \frac{V_{max}}{N}]$. Rainbow uses a categorical distribution with $M = 51$ atoms. For fair comparison, we assign $K = \lfloor \frac{M}{N} \rfloor$ atoms for the distribution of each sub-return, which results in the same network capacity as the original network architecture.

Our code is built upon dopamine framework (Castro et al. [2018]). We use the default well-tuned hyper-parameter setting in dopamine. For our updating rule in Eq. 9, we set $\lambda = 0.0001$. We run our agents for 100 epochs, each with 0.25 million of training steps and 0.125 million of evaluation steps. For evaluation, we follow common practice in Van Hasselt et al. [2016], starting the game with up to 30 no-op actions to provide random starting positions for the agent. All experiments are performed on NVIDIA Tesla V100 16GB graphics cards.

## 4.1 Comparison with Rainbow

To verify that our architecture achieves reward decomposition without degraded performance, we compare our methods with Rainbow. However we are not able to compare our method with Van Seijen et al. [2017] and Grimm and Singh [2019] since they require either pre-defined state pre-processing or specific-state resettable environments. We test our reward decomposition (RD) with 2 and 3 channels (e.g., RD(2), RD(3)). The results are shown in Figure 2. We found that our methods perform significantly better than Rainbow on the environments that we tested. This implies that our distributional reward decomposition method can help accelerate the learning process. We also discover that on some environments, RD(3) performs better than RD(2) while in the rest the two have similar performance. We conjecture that this is due to the intrinsic settings of the environments. For example, in *Seaquest* and *UpNDown*, the rules are relatively complicated, so RD(3) characterizes such complex reward better. However in simple environments like *Gopher* and *Asterix*, RD(2) and RD(3) obtain similar performance, and sometimes RD(2) even outperforms RD(3).

## 4.2 Reward Decomposition Analysis

Here we use *Seaquest* to illustrate our reward decomposition. Figure 3 shows the sub-rewards obtained by taking the expectation of the LHS of Eq.5 and the original reward along an actual trajectory. We observe that while $r_1 = \mathbb{E}(R_1)$ and $r_2 = \mathbb{E}(R_2)$ basically add up to the original reward $r$, $r_1$ dominates as the submarine is close to the surface, i.e. when it rescues the divers and

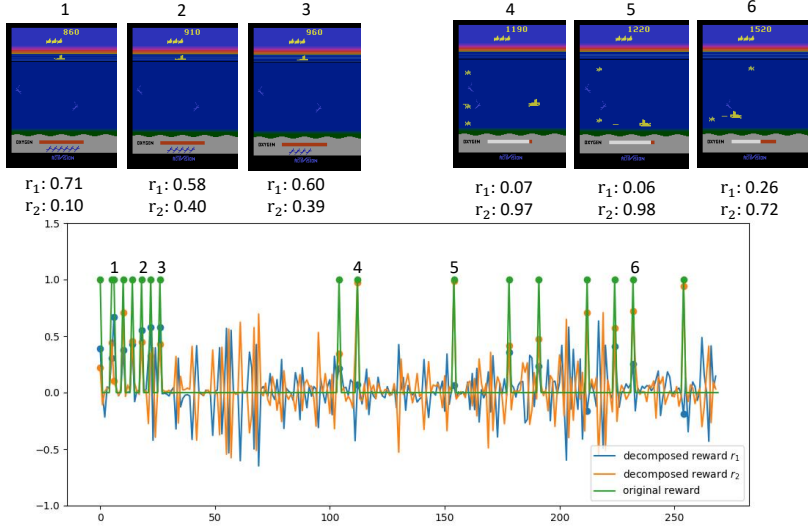

Figure 3: Reward decomposition along the trajectory. While sub-rewards $r_1$ and $r_2$ usually adds up to the original reward $r$, we see that the proportion of sub-rewards greatly depends on how the original reward is obtained.

refills oxygen. When the submarine scores by shooting sharks, $r_2$ becomes the main source of reward. We also monitor the distribution of different sub-returns when the agent is playing game. In Figure 4 (a), the submarine floats to the surface to rescue the divers and refill oxygen and $Z_1$ has higher values. While in Figure 4 (b), as the submarine dives into the sea and shoots sharks, expected values of $Z_2$ (orange) are higher than $Z_1$ (blue). This result implies that the reward decomposition indeed captures different sources of returns, in this case shooting sharks and rescuing divers/refilling oxygen. We also provide statistics on actions for quantitative analysis to support the argument. In Figure 6(a), we count the occurrence of each action obtained with $\arg\max_a \mathbb{E}(Z_1)$ and $\arg\max_a \mathbb{E}(Z_2)$ in a single trajectory, using the same policy as in Figure 4. We see that while $Z_1$ prefers going up, $Z_2$ prefers going down with fire.

### 4.3 Visualization by saliency maps

To better understand the roles of different sub-rewards, we train a DRDRL agent with two channels (N=2) and compute saliency maps (Simonyan et al. [2013]). Specifically, to visualize the salient part of the images as seen by different sub-policies, we compute the absolute value of the Jacobian $|\nabla_x Q_i(x, \arg\max_{a'} Q(x, a'))|$ for each channel. Figure 5 shows that visualization results. We find that channel 1 (red region) focuses on refilling oxygen while channel 2 (green region) pays more attention to shooting sharks as well as the positions where sharks are more likely to appear.

### 4.4 Direct Control using Induced Sub-policies

We also provide videos[2] of running sub-policies defined by $\pi_i = \arg\max_a \mathbb{E}(Z_i)$. To clarify, the sub-policies are never rolled out during training or evaluation and are only used to compute $J_{disentang}^{ij}$ in Eq. 6. We execute these sub-policies and observe its difference with the main policy $\pi = \arg\max_a \mathbb{E}(\sum_{i=1}^{M} Z_i)$ to get a better visual effect of the reward decomposition. Take *Seaquest* in Figure 6(b) as an example, two sub-policies show distinctive preference. As $Z_1$ mainly captures the reward for surviving and rescuing divers, $\pi_1$ tends to stay close to the surface. However $Z_2$ represents the return gained from shooting sharks, so $\pi_2$ appears much more aggressive than $\pi_1$. Also, without $\pi_1$ we see that $\pi_2$ dies quickly due to out of oxygen.

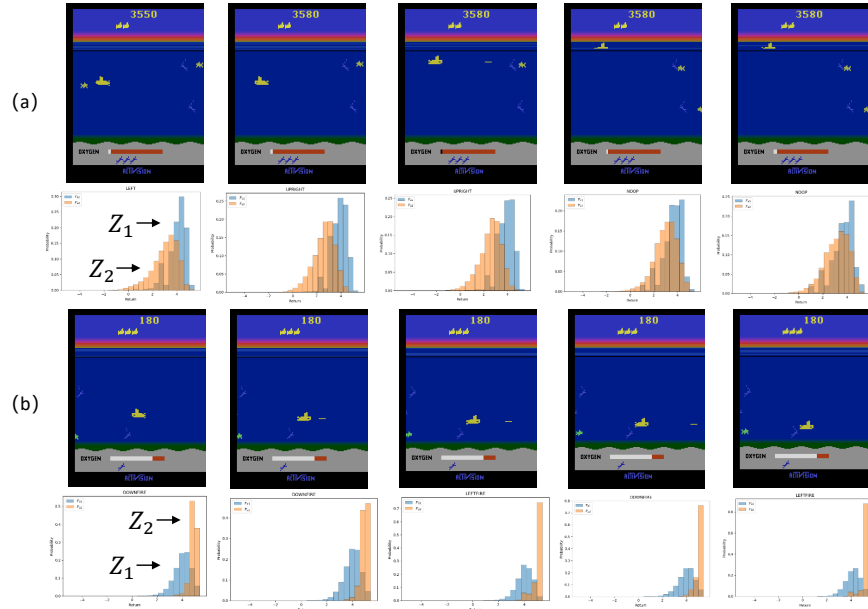

Figure 4: An illustration of how the sub-returns discriminates at different stage of the game. In figure (a), the submarine is refilling oxygen while in figure (b) the submarine is shooting sharks.

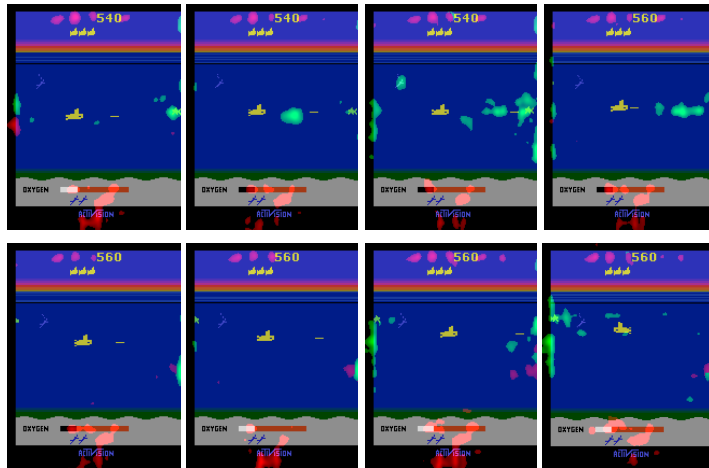

Figure 5: Sub-distribution saliency maps on the Atari game Seaquest, for a trained DRDRL of two channels (N=2). One channel learns to pay attention to the oxygen, while another channel learns to pay attention to the sharks.

## 5    Related Work

Our method is closely related to previous work of reward decomposition. Reward function decomposition has been studied among others by Russell and Zimdars [2003] and Sprague and Ballard [2003]. While these earlier works mainly focus on how to achieve optimal policy given the decomposed reward functions, there have been several recent works attempting to learn latent decomposed rewards. Van Seijen et al. [2017] construct an easy-to-learn value function by decomposing the reward function of the environment into $n$ different reward functions. To ensure the learned decomposition is non-trivial, Van Seijen et al. [2017] proposed to split a state into different pieces following domain knowledge and then feed different state pieces into each reward function branch. While such method can accelerate learning process, it always requires many pre-defined preprocessing techniques. There

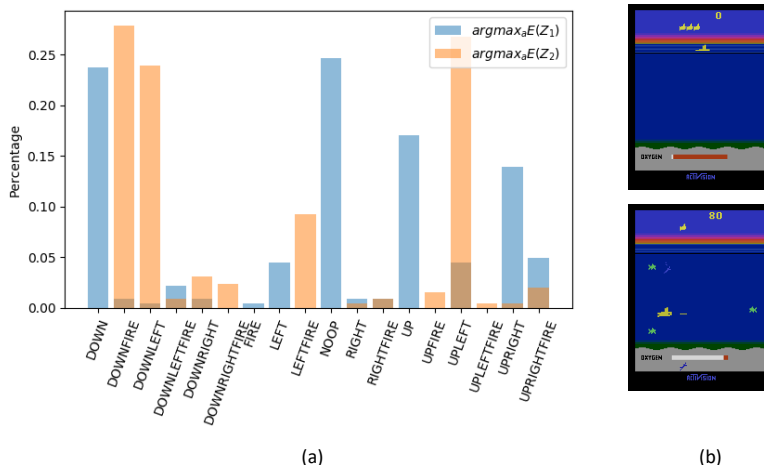

(a)                                                    (b)

Figure 6: (a) Action statistics in an example trajectory of *Seaquest*. (b) Direct controlling using two induced sub-policies $\pi_1 = \arg\max_a E(Z_1), \pi_2 = \arg\max_a E(Z_2)$: the top picture shows that $\pi_1$ prefers to stay at the top to keep agent alive; the bottom picture shows that $\pi_2$ prefers aggressive action of shooting sharks.

has been other work that explores learn reward decomposition network end-to-end. Grimm and Singh [2019] investigates how to learn independently-obtainable reward functions. While it learns interesting reward decomposition, their method requires that the environments be resettable to specific states since it needs multiple trajectories from the same starting state to compute their objective function. Besides, their method aims at learning different optimal policies for each decomposed reward function. Different from the works above, our method can learn meaningful implicit reward decomposition without any requirements on prior knowledge. Also, our method can leverage the decomposed sub-rewards to find better behaviour for a single agent.

Our work also relates to Horde (Sutton et al. [2011]). The Horde architecture consists of a large number of 'sub-agents' that learn in parallel via off-policy learning. Each demon trains a separate general value function (GVF) based on its own policy and pseudo-reward function. A pseudo-reward can be any feature-based signal that encodes useful information. The Horde architecture is focused on building up general knowledge about the world, encoded via a large number of GVFs. UVFA (Schaul et al. [2015]) extends Horde along a different direction that enables value function generalizing across different goals. Our method focuses on learning implicit reward decomposition in order to more efficiently learn a control policy.

# 6   Conclusion

In this paper, we propose Distributional Reward Decomposition for Reinforcement Learning (DR-DRL), a novel reward decomposition algorithm which captures the multiple reward channel structure under distributional setting. Our algorithm significantly outperforms state-of-the-art RL methods RAINBOW on Atari games with multiple reward channels. We also provide interesting experimental analysis to get insight for our algorithm. In the future, we might try to develop reward decomposition method based on quantile networks (Dabney et al. [2018a,b]).

### Acknowledgments

This work was supported in part by the National Key Research & Development Plan of China (grant No. 2016YFA0602200 and 2017YFA0604500), and by Center for High Performance Computing and System Simulation, Pilot National Laboratory for Marine Science and Technology (Qingdao).

## Footnotes

[2]https://sites.google.com/view/drdpaper

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
