[Supplementary Material]

## Appendix

## Non-factorial model

To show that $P(Z_i = v) = P(Z_i = v|Z_j)$ is a reasonable assumption, we also implemented a non-factorial version of DRDRL, i.e. assuming that $P(Z_i = v)! = P(Z_i = v|Z_j)$, with 3 channels. The architecture of the non-factorial model is shown in figure 1. The three sub-distributions have $K$ atoms, $K^2$ atoms and $K^3$ atoms respectively and they multiply to form a full distribution with $K^3$ atoms. To maintain similar network capacity as C51, we set $K = 4$ to form a full distribution of 64 atoms.

Figure 1: Model architecture of full distribution (non-factorial) method mentioned in section 3.1.

Ablative results of the non-factorial model are shown in the following section together with other tricks.

## Ablative Analysis

Figure 2 shows how each component of our method contributes to DRDRL. Specifically, we test the performance of '1D convolution', '1D convolution + KL', '1D-convolution + onehot' and '1D-convolution + KL + onehot' as well as the non-factorial model.

One may argue that the superior performance of DRDRL might come from the fact that fitting N simple sub-distributions with M/N categories is easier than fitting one with M categories. We include another set of experiments to justify this argument by using M atoms instead of M/N atoms for each sub-distribution.

Results show that every part of DRDRL is important. To our surprise, the KL term not only allows us to perform reward decomposition, we also observe significant differences between curves with and without the KL term. This suggests that learning decomposed reward can greatly boost the performance of RL algorithms. Together with the degraded performance of M atoms instead of M/N atoms in each sub-distribution, it is suffice to suggest that the DRDRL's success does not come from fitting easier distributions with M/N atoms.

Figure 2: Training curves.