[Reviews · NeurIPS 2019]

Reviewer 1



Originality: The work proposes an interesting framework for distributional reward decomposition. The work is not particularly novel since it is based on several prior works, albeit in a non-trivial way. For example, as discussed in point 3 in the contributions section above, the authors are inspired by [Grimm and Singh, 2019] but they propose their own disentanglement loss term. Quality:The intuitions behind the various steps seem quite reasonable to me. The experiment shows significant improvements in RL performance, and the authors provide experiments that show how the computed rewards are correlated with the ground truth. On the other hand, only decompositions of 2 or 3 terms have been performed and other decomposition methods are not assessed. Significance: Disentangled reward decomposition is a very important area in RL, so the work can be used by other researchers or practitioners. UPDATE Thanks for the authors' feedback. I acknowledge that I have read the rebuttal and the other reviews. I believe this is an interesting work that successfully brings together various promising ideas from the existing literature, so I think it is qualified for acceptance. Therefore, I keep my original score of 6. The reason why I am not giving a higher score is that the experiments are not entirely convincing to me. For instance, I raised the point about high-D experiments - the authors mentioned that their framework is in principle applicable to high-D settings but I think experiments would be needed to back up this claim. Nevertheless, I still think that this is an interesting work and relevant to the RL community.

Reviewer 2



The submission introduces a method for distributional reward decomposition which is more generally applicable than prior work, removing requirements for arbitrary resets as well as domain knowledge. The method models subrewards as Categorical distributions, treating reward composition as 1D convolution, and relying on update rules from prior work on distributional Q-learning (C51) for update rules. To further strengthen disentanglement the objective is extended to maximise the KL divergence between the distributions resulting from actions optimising for different subrewards (treating the learned Q functions as epsilon greedy policies). Overall, the work provides a valuable contribution to RL by investigating (and benefitting from) reward decomposition in a distributional setting. The combination of reward decomposition and distributional RL provides novelty and as demonstrated in the experimental section better agent performance by exploiting task structure. It would be interesting in this context to see how the approach fares in tasks with only a single source of reward and potential situations where the method might perform worse than the baseline. On the experiment section it would be important to additionally compare against distributions with the same number of atoms as it might be simply easier to fit distributions with M/N atoms than with M atoms, leading to an unplanned benefit of the proposed algorithm. In Figure 3, it would be interesting to investigate situations when either subreward spikes but not the original one to better understand the model. To be fair to prior work and provide a more complete evaluations, it would be good to compare against Van Seijen et al and Grimm and Singh in environments where their requirements are fulfilled. Minor: Couple of spelling/grammatical mistakes in lines 10, 89, 90, 101 Statement about stochastic policies could leave out the epsilon greedy part to be more general as stochasticity can also apply to continuous action policies After introducing the UVFA like trick in 3.4 is the state splitting still required? Figure 1a: To be more self-contained it would be beneficial to also explain the symbols here Possibly related work: RUDDER: Return Decomposition for Delayed Rewards; Jose A. Arjona-Medina, Michael Gillhofer, Michael Widrich, Thomas Unterthiner, Johannes Brandstetter, Sepp Hochreiter I appreciate the author feedback in particular with the additional ablation and investigation of the model which will help answer some open questions. In addition to this, I hope the authors will spend time on the promised detailed evaluation of failure cases and investigation why the approach improves performance even when its assumptions seem broken (one reward source).

Reviewer 3



I find the ideas presented in this work to be sound. Learning decomposed reward representation seems to give more representation power, which leads to improved performance. However, the ideas are quite incremental (combining two well-studied approaches), there is no new analysis and the experiments are not too impressing. To be more convinced, I would like to see ablative analysis of the results - how each component in this work (new loss, new architecture, distributional rl) contributed towards the final solution. Detailed comments: While many reward decomposition papers try to learn a different policy for each component, to be combined later on, here the focus is on learning better representations by using the decomposed representation. I would like the authors to emphasize that more in the text and explain why they took this approach. Section 3.1. This section is quite confusing. Equations are derived, but then it is explained that they are ignored. The authors mention that they performed experiments with the full distribution method (the none factorial) but they did not perform well. I would like to see the distributional model developed for this case as well as the supporting experiments to be convinced as well. The fact that the sum of two random variables is given by convolution is true for any two random variables (this is taught in basic probability courses). The way it is presented here may confuse the reader to think that there is some novelty here. I would like to see a derivation of equation 4, I am not confident that the optimal bellman equation is linear. Section 3.3. Is the projected Bellman loss novel to this work or was it proposed in the Bellamere et al (2017) paper? please be specific. If it was proposed before, then why wasn't it implemented in the Rainbow architecture? is this new loss only improves results when combined with the reward decomposition? I would like to see more experiments about this loss, with an without other components, as well as a detailed explanation regarding when was it first proposed and when was it used. In equation 7 there is a subscript i, but it is not used in equation 8, can you please explain how you move between these equations? Experiments. How were the hyperparameters selected? how are they different than the classic parameters of Rainbow? did you find the algorithm to be sensitive to these parameters? The regularized \lambda seems to be quite small. Can you elaborate on that? was it needed at all? --------------------------------------- Following the rebuttal: I appreciate the authors' effort in addressing my questions. I also find the additional experiments provided in the rebuttal to be interesting, and I believe that they improve the quality of the paper significantly. Since the authors also agreed to address most of my concerns in the final version of the paper, I am increasing my score from 4 to 6.

[Author Response · NeurIPS 2019]



Figure 1: (a) training curves. (b) saliency maps. (c) model of full distribution (non-factorial) method.

*General Response:* Thanks a lot for your comments and suggestions! We will fix typos, add more details and re-organize
the paper to improve clarity. Figure 1 shows the additional experiments. (**a**) *Ablative results to show the contribution of*
*each component.* Meanwhile, we provide the results of non-factorial method in eq.3 as well as comparison with using
M atoms for each distribution. (**b**) *Visualization using saliency maps.* We compute the absolute value of the Jacobian
$|\nabla_x Q_i(x, \arg\max_{a'} Q(x, a'))|$ for each channel. We find that channel 1 (red region) focuses on refilling oxygen while
channel 2 (green region) pays more attention to shoot sharks as well as the position where the sharks always appear. (**c**)
*The model of non-factorial method.* (**d**) *Comparison with prior works.* As stated in our paper, most prior works on
reward decomposition require either domain knowledge (Van Seijen et al. [2017]) or special environment (Grimm and
Singh [2019] requires resettable environment). So it is impossible to fairly compare our algorithm which works on
usual RL environments with theirs. Also the reported performances of those two works are quite limited: the reported
performance of Grimm and Singh [2019] is only slightly better than DQN or even lower on some games, and the work
of Van Seijen et al. [2017] only reports its performance on MsPacman. At the moment, it is difficult to reproduce
results of Van Seijen et al. [2017] and Grimm and Singh [2019] since they need either domain knowledge or resettable
environment. We will try our best to implement these prior methods and add all results in the new version.

*To Reviewer 1:* **Q1**: *Scale to high-D decomposition?* **A1**: In principle, our method can scale to high-D decomposition,
and will work well as long as the reward in the environment can be decomposed to multiple (high-D) sources of
sub-reward. In experiments we adopt only 2- or 3- decomposition because in most Atari Games the reward can be
decomposed to 2 or 3 sources of sub-reward. **Q2**: *Comparison with prior work?* **A2**: Please refer to (**d**) in general
response. **Q3**: *Why instantaneous reward rather than additive discounted reward in disentanglement loss?* **A3**: Note
that $\mathcal{F}$ in loss 6 is the distribution of sub-returns instead of sub-rewards, so our disentanglement loss consists with that
of Grimm and Singh [2019] in using additive discounted reward. The notations here might be a bit misleading; we will
refine and make them clearer.

*To Reviewer 2:* **Q1**: *Using M atoms?* **A1**: As shown in Figure 1, the performance of using M atoms for each distribution
is similar to (or slightly better than) results of using M/N atoms, which is due to larger network capacity. This suggests
that the effectiveness of our method does not come from simply fitting an easier distribution with M/N atoms. **Q2**:
*About state splitting.* **A2**: We do not use state splitting after using the one-hot embedding. **Q3**: *Situations when either*
*sub-reward spikes but not the original one.* **A3**: We looked into those situations and found that the spikes are mainly
due to over-estimations on Q. For example, returning large Q value when shark and submarine are on the same level
instead of before the shark is going to get hit by projectiles. But this makes sense because the original sub-reward is a
random variable and as stated in line 111-112, the pseudo reward is only used for visualization. **Q4**: *Failure cases with*
*a single source of reward.* **A4**: We will run experiments on more Atari games and collect negative cases to see if there
is anything in common. StarGunner in our paper is a game with a single source of reward; the experiments in Section 4
show that our method still out-performs Rainbow. We surprisingly found that in this game, one channel focuses on
upper region of state while another channel focuses on lower region of state. We will include these results in the new
version.

*To Reviewer 3:* **Q1**: *Ablative Analysis/Visualization using saliency maps/non factorial model.* **A1**: Please refer to
(**a**)/(**b**)/(**c**) in general response. **Q2**: *Why took this approach?* **A2**: While learning a different policy for each component
is intuitive, it is hard to combine those policies. Our approach uses a single policy and reaches even better score than
Rainbow. The challenge of using a single policy is that it makes reward decomposition implicit and is hard to obtain
meaningful sub-rewards; therefore, we derive a disentanglement loss to make the implicit sub-rewards as disentangled
as possible. **Q3**: *Derivation of equation 4.* **A3**: The way in which we presented equation 4 might be a bit misleading;
actually it is not difficult to derive the equation if we combine it with the distributional Bellman operator presented in
Section 2.2. Due to space limitations, we cannot include the derivation here; we will add it into the new version. **Q4**:
*Novelty of the projected Bellman loss.* **A4**: It was proposed in Bellemare et al. [2017], and implemented in the Rainbow
architecture for distributional RL. In section 3.3. We combine our proposed KL loss with projected Bellman loss to
form a final objective function. We will explicitly point out this in the new version. **Q5**: *Subscript i in eq. 7&8.* **A5**: The
subscript in eq.7 refers to the probability of the i-th atom of the distribution; in eq.8 the full distribution (all atoms) is
used to compute KL divergence, and so the subscript is omitted. **Q6**: *Hyper-parameters.* **A6**: We follow all the classic
parameters of Rainbow except the hyper-parameters that induced by our method (number of channels N, atom number
of each distribution and learning rate $\lambda$ for KL loss). For the atom number, we choose M/N in order to keep the same
network capacity as Rainbow. For $\lambda$, since KL loss is a regularization term, setting $\lambda$ as a large value (e.g. 1 or 0.1) will
hurt the agent performance. Therefore, we choose a relatively small value 0.001 and this learning rate is not sensitive.

[Meta-Review · NeurIPS 2019]

The reviewers enjoyed the paper, although expressed some concerns regarding the novelty (it combines a number of existing ideas). Still, the combination does result in a clear performance increase on a small set of Atari 2600 games. In the discussion the reviewers appreciated the additional experiments provided in the rebuttal, and reiterated the need for the final version of this paper to incorporate these and to be cleaned up. I also want to encourage the authors to report the performance of their algorithm on a larger number of Atari 2600 games -- in particular, how were these 6 games selected? Was there an unconscious bias in this selection?